 SHORT REPORT

# Mixed synapses reconcile violations of the size principle in zebrafish spinal cord

Evdokia Menelaou, Sandeep Kishore, David L McLean*

Department of Neurobiology, Northwestern University, Evanston, United States

**Abstract** Mixed electrical-chemical synapses potentially complicate electrophysiological interpretations of neuronal excitability and connectivity. Here, we disentangle the impact of mixed synapses within the spinal locomotor circuitry of larval zebrafish. We demonstrate that soma size is not linked to input resistance for interneurons, contrary to the biophysical predictions of the 'size principle' for motor neurons. Next, we show that time constants are faster, excitatory currents stronger, and mixed potentials larger in lower resistance neurons, linking mixed synapse density to resting excitability. Using a computational model, we verify the impact of weighted electrical synapses on membrane properties, synaptic integration and the low-pass filtering and distribution of coupling potentials. We conclude differences in mixed synapse density can contribute to excitability underestimations and connectivity overestimations. The contribution of mixed synaptic inputs to resting excitability helps explain 'violations' of the size principle, where neuron size, resistance and recruitment order are unrelated.

## Editor's evaluation

This is a short but incisive report on the biophysical basis of the "size principle" – an old hypothesis to explain the orderly recruitment of interneurons and motorneurons according to the size of their motor pool. The authors examine the contribution of electrical gap junctions to these recruitment phenomena in the spinal locomotor circuits in larval zebrafish. They show that the prevalence of mixed electrical/chemical synapses helps resolve some paradoxical deviations from the size principle. This is an important idea and is based on a convincing analysis of a large electrophysiology dataset acquired over many years.

*For correspondence:
david-mclean@northwestern.edu

**Competing interest:** The authors declare that no competing interests exist.

## Introduction

Neurons come in a variety of shapes and sizes, which impacts their ability to get excited. A common electrophysiological test of neuronal excitability is the measurement of input resistance. Input resistance is assessed by somatic current injection and serves to predict how easily neurons depolarize in response to synaptic current. Lower-resistance neurons are considered less excitable since larger synaptic currents are required to depolarize from rest to threshold, following Ohm's law (voltage=current×resistance). Since measurements are performed at resting potentials, the primary determinant of conductance reporting input resistance is assumed to be membrane leak channels, although voltage-dependent channels and synaptic noise can also contribute to input resistance (*Morales et al., 1987*; *Paré et al., 1998*; *Picton et al., 2018*; *Yuan et al., 2005*).

In addition to electrical or chemical synaptic excitation, mixed electrical-chemical synapses form gap junctions and release glutamate to recruit neurons (*Nagy et al., 2019*). Gap junctions provide a rapid, reliable source of depolarizing current and can enhance glutamate release at co-active mixed synapses (*Alcamí and Pereda, 2019*; *Connors, 2017*). While higher-density electrical synapses would more easily excite target neurons, at resting potentials gap junctions also provide a source of leak that decreases input resistance and creates parallel paths for current spread (*Marder et al., 2017*).

In addition, since electrophysiological assessments of connectivity often rely on spike-triggered averages to reveal postsynaptic potentials (*Mendell and Henneman, 1968*), it could be challenging to disentangle the contribution of monosynaptic currents from coupling potentials propagating indirectly through electrical synapses (*García-Pérez et al., 2004*; *Korn et al., 1973*). Consequently, the existence of mixed synapses has the potential to not only confound electrophysiological assessments of neuronal excitability, but also connectivity.

Here, we have explored this possibility within the spinal locomotor circuitry of larval zebrafish. Mixed synapses formed by reticulospinal and propriospinal interneurons provide sources of excitation during locomotion in larval zebrafish (*Bhatt et al., 2007*; *Kimura et al., 2006*; *McLean et al., 2008*; *Menelaou and McLean, 2019*; *Pujala and Koyama, 2019*; *Wang and McLean, 2014*). They also appear early in zebrafish embryos (*Miller et al., 2015*; *Saint-Amant and Drapeau, 2001*) and persist into adulthood (*Pallucchi et al., 2022*; *Song et al., 2016*). Thus, it is likely that gap junctions in mixed synapses impact electrophysiological assessments in zebrafish of all ages.

## Results and discussion

Our previous studies of spinal motor neurons in larval zebrafish have identified slow, intermediate, and fast motor units recruited during faster swimming in order of size and input resistance (*Bello-Rojas et al., 2019*; *McLean et al., 2007*; *Menelaou and McLean, 2012*). This observation in fish is consistent with the 'size principle' originally formulated in felines (*Henneman et al., 1965*) and more recently observed in flies (*Azevedo et al., 2020*). In this scenario, larger neurons are less excitable at rest due to a greater number of membrane leak channels. However, among populations of premotor spinal interneurons that regulate motor neuron recruitment, the relationship between size, input resistance, and recruitment order is more complicated. For instance, soma size (*McLean et al., 2007*) and input resistance (*Menelaou and McLean, 2019*) are not always predictive of interneuron recruitment order, 'violating' the biophysical predictions of the size principle.

Mixed synapses could explain these discrepancies, assuming input resistance measures reflect not only interneuron size but also the specificity and density of gap junctions. To test this idea, we explored potential links between size, excitability, and connectivity among molecularly defined premotor excitatory and inhibitory interneurons that coordinate the head-tail propagation and left-right alternation of cyclical body bends during swimming at different speeds (*Figure 1a and b*). The data set included unpublished recordings (n=46) and new analysis of published recordings from excitatory *chx10*-labeled V2a interneurons (*Menelaou and McLean, 2019*), and inhibitory *dbx1*-labeled V0d interneurons (*Menelaou and McLean, 2019*) and *dmrt3a*-labeled dI6 interneurons (*Kishore et al., 2020*).

Analysis of two distinct morphological classes of excitatory V2a neurons with either descending (V2a-D) or bifurcating (V2a-B) axon trajectories revealed largely overlapping sizes, but distinct input resistance values (*Figure 1c and d*, left). For inhibitory dI6 and V0d interneurons the opposite was true—neurons had largely overlapping input resistances, but distinct sizes (*Figure 1d*, middle). Within V2a subtypes, we found no significant relationship between size and input resistance, although lower resistance V2a-B neurons were largest (*Figure 1d*, left). As a result, combining the two subtypes generated a significant relationship within the V2a population ($\rho_{(248)}$=–0.61, p<0.001, n=250), albeit a weaker one compared to motor neurons (*Figure 1d*, right). For dI6 neurons, there was an even weaker correlation and no significant correlation for V0d neurons (*Figure 1d*, middle). These observations suggest size and membrane leak channels are not the exclusive arbiters of input resistance among interneurons.

Next, if mixed synapses were contributing to measurements of input resistance, we would expect lower membrane time constants in lower resistance neurons (*Getting, 1974*). Theoretically, increases in size should not impact time constants, following $\tau$=resistance×capacitance, however gap junctions can provide higher density leak conductances that impact membrane resistance more than capacitance. Consistent with this idea, membrane time constants were significantly correlated with input resistance within and between interneuron populations, reaching 25 ms in the highest resistance neurons (*Figure 1e*, left, middle). The same relationship is observed in motor neurons one synapse downstream (*Wang and McLean, 2014*), however values never exceeded 5 ms (*Figure 1e*, right). Thus, membrane time constants better predict input resistance than soma size for motor neurons and interneurons, consistent with the idea that mixed synapse density is contributing to measurements of resting excitability.

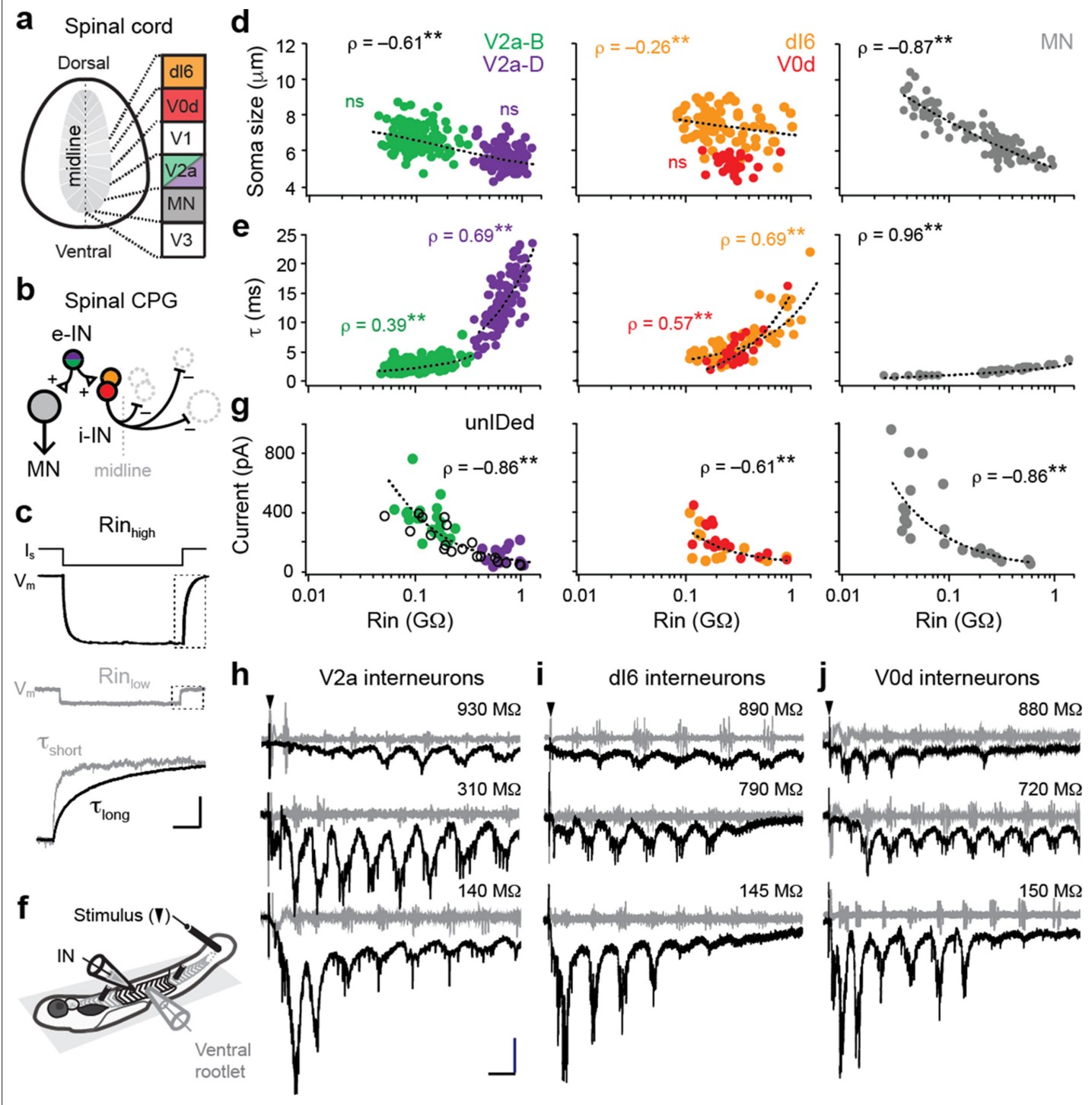

**Figure 1.** Size-independent scaling of time constants and excitation with interneuron input resistance. (**a**) Cross-section of spinal cord denoting canonical classes of molecularly defined locomotor-related interneurons. (**b**) Wiring diagram of swimming circuitry comprised of excitatory interneurons (e-IN) that provide local excitation (+) of motor neurons (MNs) and inhibitory (i-IN) interneurons, which cross the midline and silence neurons on the opposite side (–). Corresponding interneuron classes color coded as in **Figure 1a**. (**c**) Whole-cell current clamp recordings illustrate current step ($I_s$) and membrane potential deflections ($V_m$) in high and low input resistance (R) dI6 interneurons. Expanded traces below from boxed regions are normalized to illustrate differences in time constant ($\tau$) related to resistance. Scale bar, 20 pA, 10 mV, 100 ms (top), and 10 ms (expanded). (**d**) Quantification of soma size versus input resistance for motor neurons and interneurons. Significant correlations are fit with logarithmic trendlines for illustrative purposes. ns, not significant. **, significant correlation following non-parametric Spearman Rank test. V2a-D=V2a neurons with descending axons; V2a-B=V2a neurons with bifurcating axons. V2a-B, $\rho_{(143)}=-0.08$, p=0.326, n=145; V2a-D, $\rho_{(103)}=-0.14$, p=0.219, n=105; dI6, $\rho_{(70)}=-0.26$, p<0.05, n=72; V0d, $\rho_{(26)}=-0.09$,

*Figure 1 continued on next page*

*Figure 1 continued*

p=0.641, n=28; MN, $\rho_{(104)}$=–0.87, p<0.001, n=106. Source data are reported in *Figure 1—source data 1*. (**e**) Quantification of membrane time constant versus input resistance. V2a-B, $\rho$ (152)=0.39, p<0.001, n=154; V2a-D, $\rho$ (103)=0.69, p<0.001, n=105; dI6, $\rho$ (68)=0.69, p<0.001, n=70; V0d, $\rho$ (26)=0.57, p<0.001, n=28; MN, $\rho$ (39)=0.96, p<0.001, n=41. Source data are reported in *Figure 1—source data 1*. (**f**) Schematic of the recording set up for 'fictive' swimming, evoked by a brief electrical stimulus to the tail skin (black triangle), with simultaneous recordings of interneuron (IN) activity and motor output from the ventral rootlet in chemically-immobilized larvae (see Materials and methods for details). (**g**) Quantification of peak inward excitatory currents versus input resistance during 'fictive' swimming.; V2a-B, $\rho_{(22)}$=–0.58, p<0.01, n=24; V2a-D, $\rho_{(12)}$=–0.11, p=0.702, n=14; V2a-unIDed, $\rho_{(17)}$=–0.92, p<0.001, n=19; V2a-pooled, $\rho_{(55)}$=–0.86, p<0.001, n=57; dI6, $\rho_{(11)}$=–0.44, p=0.133, n=13; V0d, $\rho_{(15)}$=–0.74, p<0.001, n=17; dI6-V0d-pooled, $\rho_{(28)}$=–0.61, p<0.001, n=30. *Figure 1g* has been adapted from Figure 2i from *Kishore et al., 2014*, distributed under the terms of a Creative Commons Attribution-Noncommercial-Share Alike 3.0 Unported License CC BY-NC-SA 3.0 (https://creativecommons.org/licenses/by-nc-sa/3.0/). It is not covered by the CC-BY 4.0 license and further reproduction of this panel would need to follow the terms of the CC BY-NC-SA 3.0 license. Source data are reported in *Figure 1—source data 1*. (**h**) Whole-cell voltage-clamp recordings of V2a interneurons at calculated chloride ion reversal potential (–65 mV) with simultaneous ventral rootlet recordings (gray) reveal rhythmic inward excitatory currents (black) driving 'fictive' swimming after a brief electrical stimulus to the skin (at black arrow). Input resistance values accompany respective traces. Scale bar, 50 pA, 25 ms. (**i**) As in (**h**) but for dI6 interneurons. (**j**) As in (**h**) but for V0d interneurons.

The online version of this article includes the following source data for figure 1:

**Source data 1.** Source data from panels d, e and g in *Figure 1*.

To further test this idea, we examined cyclical excitatory drive during 'fictive' escape swimming in response to a brief electrical stimulus (*Figure 1f*), which includes mixed inputs from reticulospinal and propriospinal sources (*Menelaou and McLean, 2019*; *Pujala and Koyama, 2019*; *Wang and McLean, 2014*). As expected from components of swimming locomotor circuitry, all interneuron populations received oscillatory excitatory drive at a range of cyclical swimming frequencies (*Figure 1h–j*). Peaks in maximum oscillatory drive could occur at different frequencies among higher resistance interneurons—for example, at lower frequencies that occur near the end of the swim episode for V2a neurons (*Figure 1h*, top) or at higher frequencies near the beginning for V0d neurons (*Figure 1j*, top). This is consistent with differences in synaptic-specificity, rather than input resistance, dictating interneuron recruitment probability at different frequencies (*Ampatzis et al., 2014*).

Within and between interneuron populations, the lowest resistance neurons received the largest amount of excitatory current (*Figure 1g*). Analysis of V2a neurons revealed a significant correlation in bifurcating and morphologically unidentified V2a neurons that was not apparent in descending V2a neurons (*Figure 1g*, left). Similarly, there was a significant correlation in V0d neurons that was not observed in dI6 neurons (*Figure 1g*, middle). Notably, V0d neurons are recruited more reliably at higher frequencies compared to dI6 neurons, which operate over a broader frequency range (*Kishore et al., 2020*; *Satou et al., 2020*). Pooled excitatory and inhibitory interneuron analysis closely resembled the relationship between excitatory drive and input resistance observed in motor neurons (*Kishore et al., 2014*), where peak levels are higher commensurate with larger sizes (*Figure 1g*, right). These data suggest that higher-density mixed synapses to spinal neurons recruited preferentially during high frequency escape swimming contribute to lower input resistances.

To test this idea more directly, we performed a series of analyses on previous recordings of synaptic connectivity originating from V2a neurons recruited specifically at high swimming frequencies (*Menelaou and McLean, 2019*). We have previously shown that larger, lower resistance bifurcating V2a neurons form predominantly mixed synapses with interneurons and motor neurons, characterized by an early electrical component and a later glutamatergic component (*Figure 2a*), which are sensitive to the AMPA receptor blocker NBQX and the gap junction blocker 18-β-glycyrrhetinic acid (18-βGA), respectively (*Menelaou and McLean, 2019*). Higher resistance descending V2a neurons target the same neurons, but use predominantly glutamatergic synapses (*Figure 1b*), characterized by a faster, larger NBQX-sensitive component and a slower, lower amplitude 18-βGA-sensitive component most obvious during failures (*Menelaou and McLean, 2019*).

If mixed synapses are contributing to input resistance, we would expect larger PSPs in lower resistance neurons. On the other hand, if membrane leak channels are the primary determinant of shunt, we would expect smaller PSPs. When we plotted the amplitude of the electrical component of mixed synapses against input resistance, we found a significant negative correlation—larger amplitude

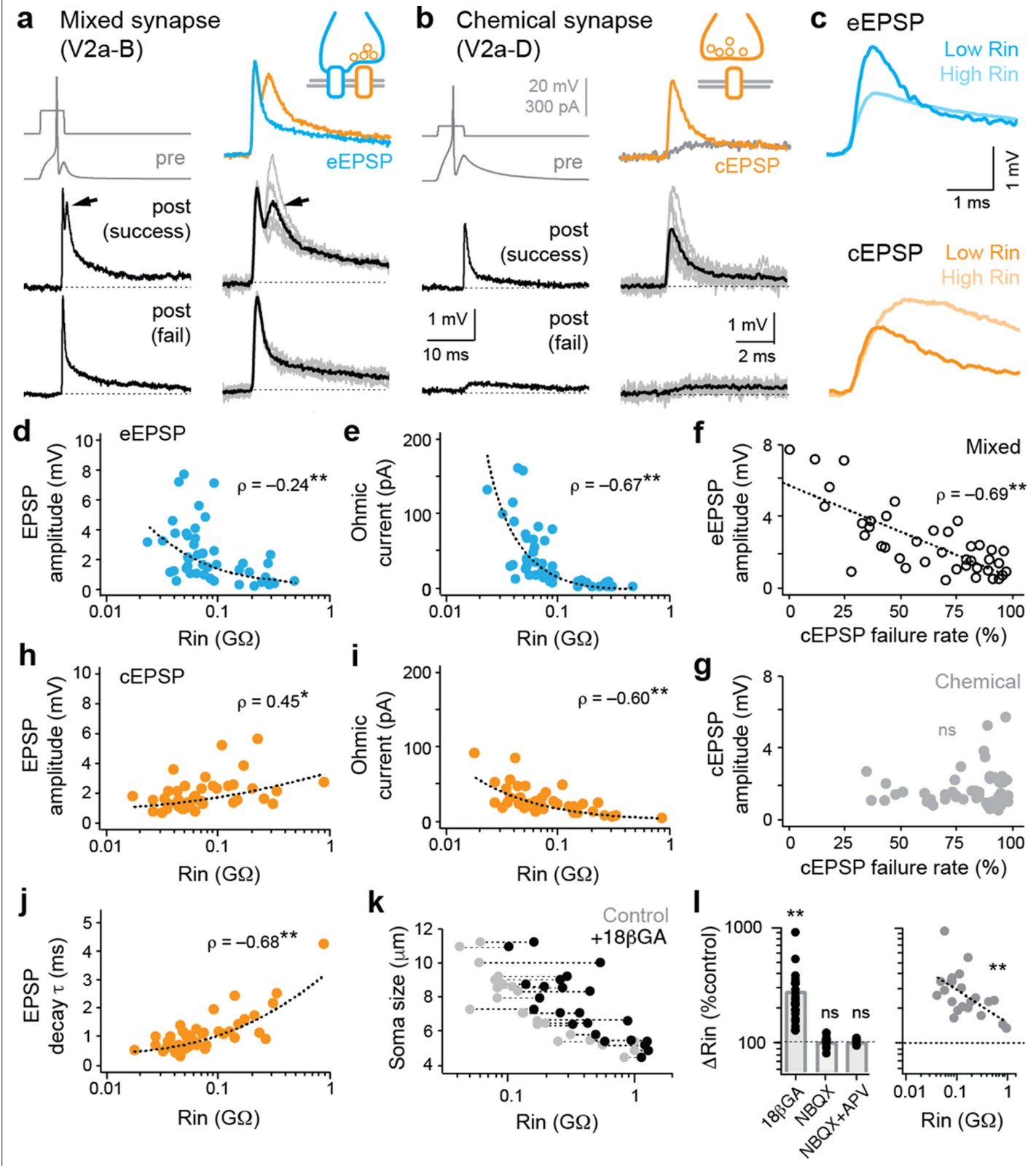

**Figure 2.** Electrical and chemical postsynaptic potentials are differentially scaled to input resistance. (**a**) Example of a mixed synaptic connection from a V2a-B neuron on slower (left) and faster (right) time scales. A brief current pulse (pre, gray traces at top left) evoked a mixed synaptic response in a postsynaptic neuron (post, black traces at bottom left). Mixed synapses are characterized by an earlier electrical component and a later, less reliable chemical component prone to failures (at arrows; fail). Averages of two distinct components of mixed synapses (electrical, blue; chemical, orange)

*Figure 2 continued*

are illustrated top right. Averages in black illustrated bottom right are superimposed on individual sweeps in gray. (**b**) As in panel (**a**), but an example of a chemical connection arising from a V2a-D neuron. Note a slow excitatory postsynaptic potential that arises via indirect electrical coupling can be resolved when the chemical connection fails. Here, the chemical connection (orange) is superimposed on the slower postsynaptic potential (gray) observed during failures. (**c**) Superimposed data illustrate differences in the amplitude and time course of electrical (blue, eEPSP) and chemical (orange, cEPSP) excitatory postsynaptic potentials related to input resistance (Rin). (**d**) Quantification of electrical EPSP amplitude versus input resistance. **, significant correlation following non-parametric Spearman rank test ($\rho_{(82)}$=–0.24, p<0.05, n=84). Source data are reported in *Figure 2—source data 1*. (**e**) Quantification of 'ohmic current' calculated from eEPSP amplitude and Rin ($\rho_{(82)}$=–0.67, p<0.001, n=84). Source data are reported in *Figure 2—source data 1*. (**f**) Quantification of the failure rate of the chemical component versus the amplitude of the electrical component of mixed eEPSPs (black open circles). **, significant following non-parametric Spearman rank test ($\rho_{(38)}$=–0.69, p<0.001, n=40). Source data are reported in *Figure 2—source data 1*. (**g**) As in (**f**), but for purely chemical synapse amplitude (ns; $\rho_{(45)}$=0.08, p=0.599, n=47). Source data are reported in *Figure 2—source data 1*. (**h**) Quantification of chemical EPSP amplitude versus input resistance. **, significant correlation following non-parametric Spearman rank test ($\rho_{(41)}$=0.45, p<0.01, n=43). Source data are reported in *Figure 2—source data 1*. (**i**), Quantification of 'ohmic current' calculated from cEPSP amplitude and Rin ($\rho_{(41)}$=–0.60, p<0.001, n=43). Source data are reported in *Figure 2—source data 1*. (**j**) Quantification of chemical EPSP decay times. **, significant correlation following non-parametric Spearman rank test ($\rho_{(41)}$=0.68, p<0.001, n=43). Source data are reported in *Figure 2—source data 1*. (**k**) Quantification of soma size versus Rin before (gray) and after (black) 18βGA application, illustrating the increase in resistance values regardless of size. Dotted lines link the same neurons (n=11 motor neurons, 12 interneurons). Source data are reported in *Figure 2—source data 1*. (**l**) Left, quantification of input resistance change expressed as a percent of controls in the presence of the gap junction blocker 18βGA versus glutamatergic blockers NBQX and/or APV. **, significant difference following non-parametric Mann-Whitney U-test (18βGA; $U_{(22)}$=0, p<0.001, n=23). ns, not significant (NBQX; $U_{(13)}$=84, p=0.520, n=14; NBQX+APV; $U_{(4)}$=10, p=0.602, n=5). Right, quantification of the percentage increase in input resistance in the presence of 18βGA as a function of initial input resistance. **, significant correlation ($\rho_{(21)}$=–0.61, p<0.01, n=23). Source data are reported in *Figure 2—source data 1*.

The online version of this article includes the following source data for figure 2:

**Source data 1.** Source data for panels d-l in *Figure 2*.

electrical excitatory postsynaptic potentials (eEPSPs) are observed in lower resistance neurons (*Figure 2c and d*). Higher-density eEPSPs to lower resistance neurons were also reflected in the 'Ohmic' current calculated using V=IR (*Figure 2e*), with individual values exceeding 100 pA. In addition, larger amplitude eEPSPs were linked to lower failure rates of the chemical component (*Figure 2f*). This is consistent with enhanced synchronous chemical excitation by depolarization spreading through the gap junctions of co-active mixed synapses (*Liu et al., 2020*; *Pereda et al., 2004*; *Song et al., 2016*). No link to failure rates was observed for purely chemical EPSPs (cEPSPs), despite a similar range of amplitudes (*Figure 2g*).

Next, we took advantage of the fact that descending V2a neurons can target the same interneurons and motor neurons with purely glutamatergic synapses to see if excitation from chemical EPSPs would be weaker in lower resistance neurons at rest, per the biophysical predictions of Ohm's law. Consistent with previous studies (*Burke, 1968*; *Mendell and Henneman, 1971*), the amplitude of chemical EPSPs was positively correlated with input resistance (*Figure 2h*), giving the impression that lower resistance neurons are less excitable, despite increases in Ohmic current up to 100 pA that could accommodate for decreased excitability (*Figure 2i*). We also observed a positive correlation between chemical EPSP decay time constant and input resistance (*Figure 2j*), as expected if mixed synapses are contributing to resistance measures and impacting integrative properties.

Thus far, the data suggest that gap junctions within mixed synapses are contributing to measurements of resting excitability in the zebrafish spinal cord. If so, we would expect that input resistance measurements would be sensitive to 18-βGA. Analysis of resistances before and after application revealed increased values in small and large neurons (*Figure 2k*), consistent with the broad contribution of gap junctions to electrophysiological measurements of excitability. The impact on excitability was specific to electrical synapses, since blockade of AMPA receptors or both AMPA and NMDA receptors had no significant effect on input resistance (*Figure 2l*, left). Critically, the relative impact of gap junction blockade was greatest in the lowest resistance neurons (*Figure 2l*, right). This is consistent with measurements of excitatory currents (*Figure 1g and h*) and postsynaptic potentials (*Figure 2d and e*), and a larger contribution of mixed synapses to leak measurements in lower resistance neurons.

Finally, to verify if mixed synapse density alone could explain our experimental observations, we turned to a user-friendly computational model (NeuroSim5; see Materials and methods). Different-sized model somata received simulated cEPSPs from a single source with identical amplitudes and waveforms (*Figure 3a and b*), assuming synaptic scaling with size following $\tau$ =RC. However,

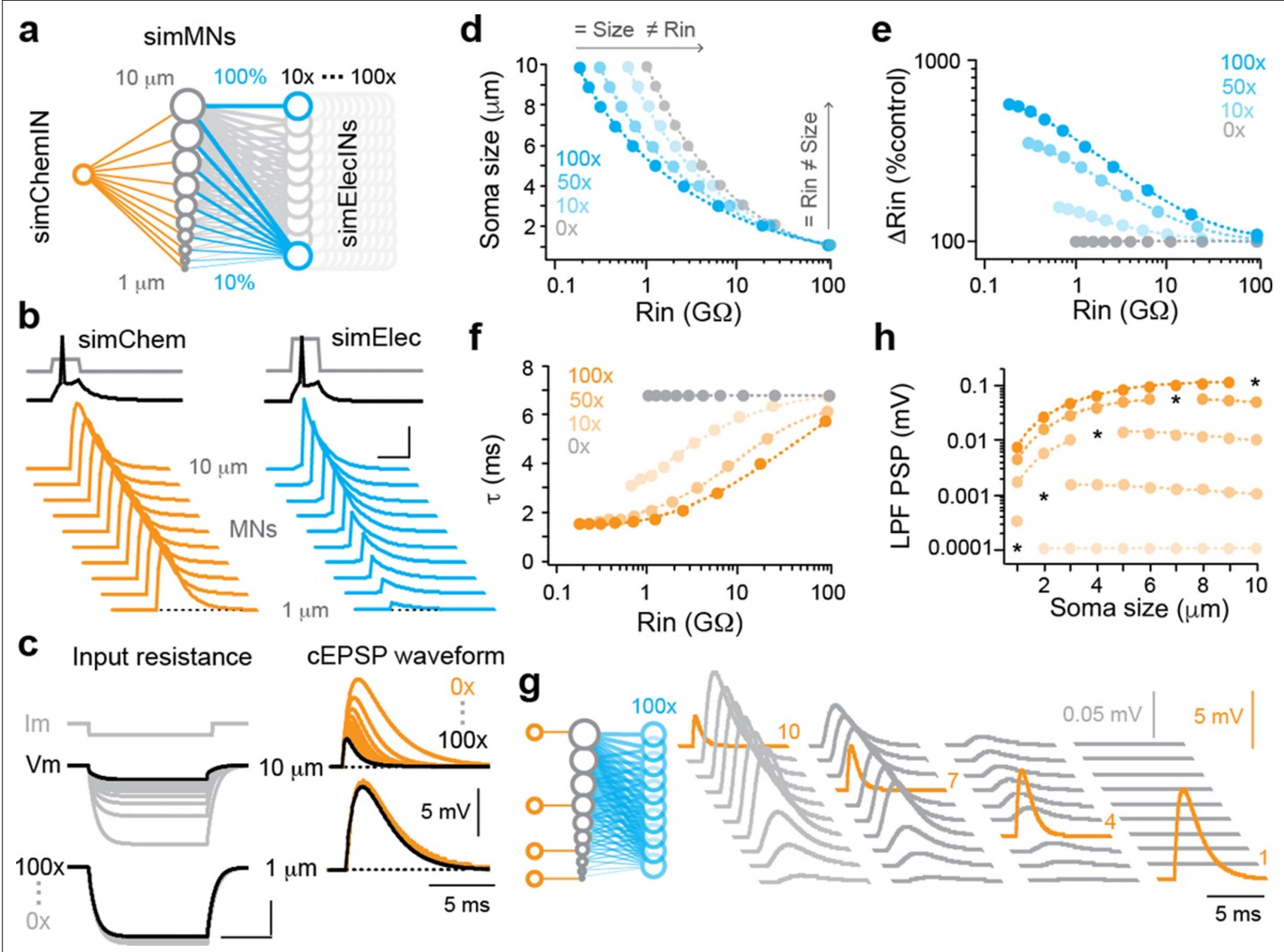

**Figure 3.** Simulations of convergent electrical and chemical synapses recapitulate experimental results. (**a**) Schematic of the simulated network. A simulated interneuron forming chemical synapses (simChemIN, orange) with 10 simulated motor neuron (sim MNs, gray) somata ranging from 1 to 10 μm in diameter. Simulated interneurons forming electrical synapses (simElecINs, blue) can vary in number from 10 to 100, with a convergence ratio ranging from 10% to 100% based on size (i.e., the largest MNs receive 100% of inputs, while the smallest receive 10%). (**b**) Simulated chemical and electrical PSPs evoked by a single spike in the respective interneurons illustrate similarities (chemical) or differences (electrical) in amplitude related to size and excitability. Note increased current pulse amplitude in the simulated electrical interneuron due to a larger size and more electrical coupling lowering resistance. Scale bar, 20 pA, 45 mV (spikes), 5 mV (PSPs), and 5 ms. (**c**) Left, simulated current injection demonstrates the impact of increasing electrical synapse convergence (from 0x to 100x) on input resistance. Scale bars, 10 pA, 2.5 mV (top), 250 mV (bottom), and 20 ms. Right, simulated chemical PSPs illustrate the impact of increased convergence on amplitude and time course. (**d**) Quantification of relationship between soma size and input resistance with increasing electrical synapse convergence (0x, 10x, 50x, and 100x). Arrows indicate axes illustrating how the same sized neurons can have different resistances or the neurons with the same resistances can be different sizes. Source data is reported in are reported in *Figure 3— source data 1*. (**e**) Quantification of the relationship between changes in Rin relative to control values with increasing electrical synapse convergence. Source data are reported in *Figure 3—source data 1*. (**f**) Quantification of the relationship between membrane time constant (t) and input resistance with increasing electrical synapse convergence. Source data are reported in *Figure 3—source data 1*. (**g**) Schematic on left illustrates that individual motor neurons receive input from only one simulated chemical interneuron, while electrical synapse convergence is at maximum value (100x). Simulated chemical PSPs and the resulting coupling potentials are illustrated to the right. (**h**) Quantification of the amplitude of the low-pass filtered (LFP) coupling postsynaptic potential versus input neuron (marked by asterisks), as in (**g**). Source data are reported in *Figure 3—source data 1*.

The online version of this article includes the following source data for figure 3:

**Source data 1.** Source data for panels d, e, f, and h in *Figure 3*.

individual electrical synapse conductances were weighted to achieve larger eEPSPs in larger neurons (*Figure 3b*), per our observations. To simulate different patterns of synaptic convergence from mixed sources related to size, the largest neurons received 100% of total inputs, while the smallest only 10% (*Figure 3a*). This approximates the tenfold difference in excitatory synapse number reported in larval zebrafish motor neurons (*Bello-Rojas et al., 2019*). By increasing the number of convergent sources, the model allowed us to explore both size-dependent and size-independent changes in resting excitability via increases in electrical synapse density.

As expected, increases in the number of convergent sources of electrical synapses led to systematic decreases in measurements of input resistance using simulated current injection (*Figure 3c*), which were most obvious in the largest, lowest resistance neurons (*Figure 3d,e*). We observed systematic decreases not only in the amplitude (*Figure 3c*), but also the time constant (*Figure 3f*) of simulated chemical PSPs, increasing the dynamic range of integration. By varying the amount of convergent sources of electrical synapses, size and input resistance could be uncoupled (*Figure 3d*). This helps explain how interneurons can be the same size with different input resistances (*Figure 1d*, left) or different sizes but the same input resistances (*Figure 1d*, middle). If mixed synapses are often used in high-frequency swimming circuitry, then differences in convergence could also explain recruitment out of order based on size and input resistance (*McLean et al., 2007*; *Menelaou and McLean, 2019*)— higher density electrical synapses lower resistance at rest independent of size, but during activity provide stronger excitation to increase recruitment probability.

One consequence of increasing the convergence of different sources of electrical synapses is the creation of parallel paths for current to spread (*Marder et al., 2017*). To examine this more closely, we simulated targeted chemical inputs to neurons of different sizes at maximum electrical synapse convergence (*Figure 3g*, left) and measured the amplitude of the coupling potentials (*Figure 3g*, right). As expected, the amplitude of the direct chemical EPSP varied as a function of size and convergence ratio (*Figure 3g*), with current distributed predominantly to neurons sharing the most afferents (*Figure 3h*). The low pass filtering provided by electrical synapses generated slower coupling potentials that were orders of magnitude lower in amplitude (*Figure 3g and h*). Simulated coupling potentials resembled slower, lower amplitude 18-βGA-sensitive potentials observed during chemical failures (*Figure 2b*, bottom) and in unconnected pairs (*Menelaou and McLean, 2019*). We and others have argued these filtered potentials reflect indirect electrical continuity through gap junctions (*García-Pérez et al., 2004*; *Korn et al., 1973*), suggesting they more accurately reflect shared afferents than direct efferents.

Collectively, our findings suggest that mixed synapses contribute to excitability underestimations in zebrafish spinal cord and help reconcile reported violations of the size principle. We also find that mixed synapses can contribute to connectivity overestimations by acting as conduits for synaptic current and confirm coupling potentials are easily distinguished based on kinetics (*García-Pérez et al., 2004*; *Menelaou and McLean, 2019*). Studies of the premotor Ia reflex pathway in cats have revealed dual and single component EPSPs with different kinetics (*Burke, 1968*; *Mendell and Henneman, 1971*). These were attributed to filtering by motor neuron dendrites and differences in the density of leak channels (*Gustafsson and Pinter, 1984*; *Rall et al., 1967*), but could also reflect axon collaterals connecting motor neuron somata via mixed synapses. In mammals, mixed synapses are found in the spinal cord (*Rash et al., 1996*), Ia circuit function is disrupted in connexin36 mutants (*Bautista et al., 2012*), and larger, faster motor units receive denser connexin36 innervation from excitatory V0 interneurons (*Recabal-Beyer et al., 2022*). So, mixed synapses could also help reconcile conflicting observations related to size, synaptic drive, and neuronal excitability in mammalian spinal cord (*Burke, 1981*; *Enoka and Stuart, 1984*; *Henneman, 1985*).

More broadly, connexin36 and vesicular glutamate transporter co-localization are observed in a variety of brain circuits optimized for processing high-frequency information reliably at speed (*Nagy et al., 2019*). The use of electrical synapses likely expands neuronal integrative properties beyond what can be achieved by leak channels and size alone (*Alcamí and Pereda, 2019*; *Galarreta and Hestrin, 2001*). This would enable rate and/or timing codes to execute orderly patterns of recruitment among targeted ensembles of motor neurons and spinal interneurons using temporal summation (*Wang and McLean, 2014*), as observed in other sensory and motor circuits (*Ainsworth et al., 2012*; *Sober et al., 2018*). In this scenario, links between input resistance and recruitment order arise by gradations in mixed synapse density and convergence among various sized neurons. This could

explain why the size principle best predicts function among motor neurons, which have the maximum capacity for synaptic convergence as the 'final common path' for all behavioral output (*Sherrington, 1906*).

## Materials and methods

### Animals

Adult zebrafish (*Danio rerio*) and their offspring were maintained at 28.5°C in an in-house facility (Pentair Aquatic Eco-Systems, Apopka, FL). All the data reported in this study was collected using 4- to 5-day-old wildtype, Tg[*chx10*:GFP] and Tg [*chx10*:lRl-GFP] (*Kimura et al., 2006*), *parg*$^{mn2Et}$ (*Balciunas et al., 2004*), Tg[*glyt2*:GFP] (*McLean et al., 2007*), Tg[*dbx1b*:cre] and Tg[*glyt2*:lRl-Gal4;UAS:GFP] (*Satou et al., 2012*), and Tg[*dmrt3a*:GFP] and Tg[*dmrt3a*:Gal4;UAS:GFP] (*Satou et al., 2020*) zebrafish larvae. At this stage, zebrafish larvae have fully inflated swim bladders and are free swimming, but have not yet sexually differentiated. All procedures conform to NIH guidelines regarding animal care and experimentation and were approved by Northwestern University Institutional Animal Care and Use Committee (Animal Study Protocol #IS00002671).

### Electrophysiology

Electrophysiological recordings from spinal motor neurons and interneurons were performed as described in *Menelaou and McLean, 2012*, *Kishore et al., 2014*, *Wang and McLean, 2014*, *Menelaou and McLean, 2019*, and *Kishore et al., 2020*. Briefly, zebrafish larvae were anesthetized in MS-222 (0.02% w/v; Western Chemical, Ferndale, WA) and then immobilized in α-bungarotoxin (0.1% w/v; MilliporeSigma, St. Louis, MO), both dissolved in extracellular solution (compositions in mmol/ l: 134 NaCl, 2.9 KCl, 1.2–2.1 MgCl2, 2.1 CaCl2, 10 HEPES, 10 glucose, adjusted to pH 7.8 with NaOH) for 5–10 min and transferred to a dish containing drug-free extracellular solution. Larvae were then secured to the elastomer-lined glass bottom and carefully dissected to expose the muscle and spinal cord using custom-edged tungsten pins and fine forceps. Whole-cell recordings from spinal neurons were performed standard wall glass capillaries with resistances between 5 and 20 MΩ and backfilled with current-clamp patch solution (compositions in mmol/l: 125–130 K-gluconate, 2–4 MgCl2, 0.2–10 EGTA, 10 HEPES, 4 Na2ATP, adjusted to pH 7.3 with KOH). Electrodes were mounted on motorized micromanipulators (Sutter Instruments, Novato, CA or Scientifica, Clarksburg, NJ) for targeting and recording. To confirm morphology, patch solution contained either Alexa Fluor 488 or 568 hydrazide (final concentration 50 μmol/l) or sulforhodamine-B acid chloride (0.025% w/v). Epifluorescent and differential contrast images were collected using a cooled Rolera-XR CCD camera (Teledyne Photometrics QImaging, Tucson, AZ) mounted on an AxioExaminer upright microscope equipped with a 40×/1.0 NA water immersion objective (Carl Zeiss, White Plains, NY). Images were captured using Qcapture Suite imaging software (QImaging) and analyzed using ImageJ (NIH, Bethesda, MD). Electrophysiological recordings were acquired using a Multiclamp 700B amplifier, a Digidata series 1322A digitizer, and pClamp software (Molecular Devices, San Jose, CA). Standard corrections for bridge balance and electrode capacitance were applied in current-clamp mode.

Excitatory currents were recorded in voltage-clamp mode (holding potential, –65 mV) using the same intracellular solution as for current clamp recordings. Values were corrected by using a calculated liquid junction potential of –11 mV. No series compensation was used and only data where the series resistance was below 60 MΩ was included in the analysis. Electrophysiological data were only included for analysis if recorded neurons had a resting membrane potential at or below –45 mV (−50) as an indication of health. Note that motor neuron voltage-clamp recordings presented in *Figure 1f* were performed using a different patch solution (composition in mmol/l: 122 CsMeSO3, 0.1–1 QX314-Cl, 1 TEA-Cl, 2 MgCl2, 4 Na2-ATP, 10 HEPES, and 1 EGTA) and series compensation was used. Whole-cell electrical signals were filtered at 30 kHz and digitized at 63–100 kHz at a gain of 10 (feedback resistor, 500 MΩ).

To simultaneously monitor 'fictive' motor activity during whole-cell recordings, a larger diameter electrode (~20–50 μm) fashioned from a patch electrode was placed over the intermyotomal cleft to record extracellularly from peripheral motor nerves. Fictive motor activity was triggered by a tungsten concentric bipolar electrode lowered onto the skin and a brief electrical stimulus (2–10 V; 0.1–0.4 ms).

Extracellular signals from the peripheral motor nerves were amplified at a gain of 1000 and digitized with low-frequency and high-frequency cutoffs set at 300 and 5000 Hz, respectively.

Connectivity was assessed by delivering 5 ms step pulses at a low frequency (<2 Hz) to elicit a single spike in the presynaptic cell while assessing postsynaptic responses in current clamp mode. For pharmacological experiments, the glutamate receptor antagonist NBQX (10 µmol/l; Abcam, Cambridge, MA) and AP5 (100 µmol/l; Abcam) were dissolved in extracellular solution and delivered to the perfusate by a gravity-fed perfusion system. The gap junctional blocker 18-beta-glycyrrhetinic acid (100–150 µmol/l; MilliporeSigma) was first dissolved in DMSO to obtain a stock solution at 200 mM and was diluted to its final concentration in extracellular solution.

## Data analysis

For soma size, the diameter was measured from DIC images using ImageJ (NIH). Motor neuron data from *Menelaou and McLean, 2012* were re-analyzed using this method in *Figure 1d*. Electrophysiological data were analyzed using Matlab (Mathworks, Natick, MA), Igor Pro 6.2 (Wavemetrics, Portland, OR) or DataView (University of St Andrews, St Andrews, Scotland) and organized in Microsoft Excel. Input resistance was determined by taking the average of at least three hyperpolarizing pulses in current-clamp mode (–10 to 50 pA) within a linear range of the current-voltage relationship. In most pharmacology experiments (n=33 out of 43), input resistance before and after drug application was calculated in voltage-clamp mode using 5 mV steps, the remainder were in current-clamp mode. Input resistance following pharmacological treatment was normalized to the control value, which was set to 100% and reported as percent of control input resistance. Synaptic currents were analyzed from at least five 'fictive' swim bouts per fish to obtain the peak excitatory current measured from baseline. To assess the membrane time constant, the decay phase of the voltage response following a square hyperpolarizing step was measured from the end of the step to baseline in at least three voltage responses and was best fit by a double exponential. The time constant ($\tau$) was then calculated from the sum of the two exponentials ($a*e^{-x*\tau 1}+b*e^{-x*\tau 2}$) and weighted time constants were calculated from the percent contribution of each component as follows: $\tau = a/(a+b)*\tau 1 + b/(a+b)*\tau 2$. Motor neuron data from *Wang and McLean, 2014* were re-analyzed using this method in *Figure 1e*.

The amplitude of the EPSP was calculated by subtracting the averaged baseline value taken 2 ms prior to presynaptic spike from the peak depolarizations of successful events. Synaptic failure rates were expressed as a percentage determined by dividing the number of failures by the total number of spike-triggered events (10–200 sweeps). In order to get an accurate measurement of the chemical component in mixed synaptic responses, the early electrical component was subtracted out. This was achieved by taking the average from electrical responses where the chemical component was absent during failures and subtracting it from all the sweeps in that experiment. For chemical postsynaptic potentials, the decay time was fit by a double exponential between peak and 20% of peak amplitude. The decay tau was measured as the first constant t1 from the fit, given the potential confound of slow electrical events. Semi-log plots and linear fits were used to illustrate significant trends in the data per *Burke, 1968*.

## Simulations

NeuroSim5 is a teaching tool that provides realistic simulations of neural activity at the cellular and small systems level (https://www.st-andrews.ac.uk/~wjh/neurosim/). The software provides an intuitive, configurable interface that enables simulation of passive membrane properties, size, and both chemical and electrical synapses. For simulations, neurons are modeled as spheres and we chose a range in diameter from 1 to 10 µm for motor neurons, 5 µm for chemical interneurons, and 7 µm for electrical interneurons, to overlap with our experimental observations. According to default settings for passive properties, membrane capacitance was set at 1 µF/cm$^2$, leak equilibrium potential and resting potential at –60 mV, and leak conductance at 0.3 mS/cm$^2$. The model used integrate-and-fire spikes with a threshold set at –40 mV, spike peak at 10 mV, spike strength at 10, relative accommodation at 0.3, accommodation tau at 10 ms, AHP conductance at 0.4 mS/cm$^2$, AHP time constant at 3 ms, AHP equilibrium potential at –70 mV, and absolute refractory period at 2 ms (https://www.st-andrews.ac.uk/~wjh/neurosim/TutorialV5_3/Implement.html#SpikingProperties). For chemical synapses, the equilibrium potential was set at 0 mV, a fixed synaptic conductance set at 0.25 mS/cm$^2$, following a single exponential with a 1 ms decay rate to match experimental observations. In the simulation,

chemical synaptic conductances are defined as normalized values and scaled internally according to the surface area of the neuron specified by its diameter. In all simulations, only a single spike was performed per run to evoke a single PSP.

Electrical synaptic conductances were non-rectifying and defined as absolute values in nS, because the synapse couples neurons with different diameters and hence different absolute membrane conductances (making coupling asymmetrical). For each simulated motor neuron, we calculated electrical synaptic conductances in nS using $y=0.01x^3$, where x equals diameter, again to match our experimental observations. In the simulation, electrical synapses are modeled as non-specific electrical coupling conductances linking the two neurons. As a result, current flows from one neuron to the other whenever there is a difference between the membrane potentials of the two neurons. The model did not include dendritic or axonal compartments or synaptic interactions between interneurons, and so cannot account for the additional filtering properties of these biological features. To model the contribution of synaptic convergence to differences in electrical synapse density, we started with 10 interneurons all 10 of which projected to the 10 μm diameter motor neuron, 9 to the 9 μm diameter, 8 to the 8 μm, and so forth, then duplicated this scheme up to 10 times totaling 100 interneurons. This approximates our experimental observations, where excitatory synapse density identified by fluorescently tagged postsynaptic density-95 protein demonstrates about 10 synapses in the smallest motor neurons and 100 in the largest (*Bello-Rojas et al., 2019*). Current pulses to evoke spikes in the simulated interneurons were 5ms in duration and 10 pA in amplitude in smaller chemical interneurons and 30 pA in larger electrical interneurons. Simulations of hyperpolarizing current injection were performed with 50 ms duration pulses 5 pA in amplitude. Data analysis was performed in simulations as described above for biological electrophysiological data.

## Statistical analysis and reporting

Before statistical analysis, all data were tested for normality and were not normally distributed, so nonparametric tests were used. Comparisons between two groups were performed using a Mann-Whitney U-test and correlations between different properties were determined using a Spearman rank test. Degrees of freedom are reported parenthetically with the respective U or rho ( $\rho$ ) values of these tests, according to convention. Statistical analysis was performed using StatPlus Professional (AnalystSoft, Alexandria, VA) in conjunction with Microsoft Excel. Significance was set at $p<0.05$ and exact p values are reported except for $p<0.001$.

## Acknowledgements

The authors thank CJ Heckman and Michael Jay for comments on the manuscript, Christopher Vaaga for advice on time constant analysis, and William Heitler for developing NeuroSim and his guidance navigating it over the years.

## Additional information

### Funding

| Funder | Grant reference number | Author |
| --- | --- | --- |
| National Institutes of Health | R21 NS125187 | David L McLean |
| National Institutes of Health | R21 NS125207 | David L McLean |
| National Institutes of Health | U19 NS104653 | David L McLean |

The funders had no role in study design, data collection and interpretation, or the decision to submit the work for publication.

## Author contributions
Evdokia Menelaou, Sandeep Kishore, Formal analysis, Investigation, Writing – original draft, Writing – review and editing; David L McLean, Conceptualization, Formal analysis, Supervision, Funding acquisition, Investigation, Visualization, Writing – original draft, Writing – review and editing

## Author ORCIDs
David L McLean (ID) http://orcid.org/0000-0001-6337-2301

## Ethics
All procedures conform to NIH guidelines regarding animal care and experimentation and were approved by Northwestern University Institutional Animal Care and Use Committee (Animal Study Protocols #IS00019359 and #IS00019319).

## Decision letter and Author response
Decision letter https://doi.org/10.7554/eLife.64063.sa1
Author response https://doi.org/10.7554/eLife.64063.sa2

# Additional files

## Supplementary files
• Transparent reporting form

## Data availability
All data generated or analyzed during this study are included in the manuscript and supporting files. Source data files are provided for Figures 1-3.

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
