## [Editor Report]

This is a short but incisive report on the biophysical basis of the "size principle" – an old hypothesis to explain the orderly recruitment of interneurons and motorneurons according to the size of their motor pool. The authors examine the contribution of electrical gap junctions to these recruitment phenomena in the spinal locomotor circuits in larval zebrafish. They show that the prevalence of mixed electrical/chemical synapses helps resolve some paradoxical deviations from the size principle. This is an important idea and is based on a convincing analysis of a large electrophysiology dataset acquired over many years.

---

## [Decision Letter]

**Decision letter after peer review:**

Thank you for submitting your article "Gap junctions impact synaptic integration and orderly recruitment in the zebrafish spinal cord" for consideration by *eLife*. Your article has been reviewed by 3 peer reviewers, including Markus Meister as Reviewing Editor and Reviewer #1, and the evaluation has been overseen by Richard Aldrich as the Senior Editor. The following individual involved in review of your submission has agreed to reveal their identity: Alberto Pereda (Reviewer #2).

The reviewers have discussed the reviews with one another and the Reviewing Editor has drafted this decision to help you prepare a revised submission.

Consensus review:

This is a short but incisive report on the biophysical basis of the "size principle" – an old hypothesis to explain the orderly recruitment of interneurons and motorneurons according to the size of their motor pool. The conventional version of this idea assumes that neurons with different size motor pools receive synaptic inputs of similar magnitude but vary in membrane resistance and thus in excitability. In this study, the authors examine the contribution of electrical gap junctions to these recruitment phenomena in the spinal locomotor circuits in larval zebrafish. The paper is based on a re-analysis of a large electrophysiology dataset they acquired over many years. The authors show that the neurons with lower input resistance and faster time constant are likely to receive more electrical synaptic input, and that the abundance of electrical synapses plays a role in reducing the input resistance of these neurons. They further suggest that the electrical synapses prioritize their own signal based on the correlations they observed across cell types, which remains to be tested experimentally.

The following suggestions for revision include: improvements in readability; clarification of experimental results; the inclusion of some controls; and an expanded discussion.

General readability and figures:

1. Please spell out early on whether you are considering electrical synapses *onto* the motor neurons, or *among* the motor neurons, or both. The question arises already in the abstract. Obviously these scenarios predict different consequences.

2. Please include one or more equivalent circuit diagrams (ligand-gated conductances, driving forces, electrical coupling) to illustrate the postulated consequences of electrical synapses in different parts of the circuit.

3. Figure 1a: This is not helpful. Hard to spot any differences between the left and right panel. What is the significance of some leak channels being green and others white? Why are there no electrical junctions between neurons anywhere?

4. Please justify the use of log-log plots in the figures? Is there a theoretical basis for the linear data fits on log-log scales?

Clarification of results:

5. Line 82: Explain better how you separated electrical and chemical components of transmission. Is it assumed that the electrical input (membrane voltage) is the same whether or not the chemical transmission fails? If so is that justified? An equivalent circuit diagram might help. Also was the separation validated by pharmacology? Blockers are only mentioned in connection with Figure 3.

6. Line 105: What is "reliability"? What motivates this measure? Why does it have units of mV? Explain how this measure should depend on the biophysical parameters of electrical/chemical transmission. Presumably the effects will differ depending on whether the electrical synapses are onto or among the target neurons. Without such explanation it is hard to follow the claim in line 108.

7. Cell types: In all of the analyses, various cell types are collapsed together, raising the possibility that the observed trend arises from particular cell types that differ from the others. To exclude this possibility, the authors could colour-code each data point in the scatter plots based on the cell type (or the combination of the cell types for Figure 2 and 3). For statistical analyses, the authors could examine the contribution of both cell-type (or combination of cell types) and explanatory variable (input resistance for Figure 1, 2 and 3c, the amplitude of PSP for Figure 2a) at the same time using a generalized linear model. The introduction could say more about the cell types examined in this study (type 1 and 2 V2a cells and their differences and similarities). The discussion could address how the *distribution* of electrical synapses among different cell types might affect spinal circuits.

8. Electrical synapses prioritize their own signals: One of the major claims of the paper is that "stronger electrical synapses further prioritize their own signals by shunting other inputs and enhancing synchronous electrical inputs by lateral excitation". This claim seems to require further analysis. To directly demonstrate that electrical synapses shunt other inputs, one might show that *other inputs* get stronger after blocking the electrical synapse. Furthermore, it is confusing that the authors rely on Menelaou and McLean 2019 to support the claim of shunting because the connections examined in this paper come from the neurons (type I and II V2a neurons) that are active at the same time (during fast swim cycles). It would be more appropriate to examine how these electrical synapses impact the inputs from the neurons active during slow swim cycles.

Relation to prior work:

9. Line 15: "Our findings challenge the view that leak channels alone dictate input resistance and membrane time constant…." In addition to leak channels, it is well known that synaptic transmission affects input resistance and membrane time constant, in particular for inhibitory synapses. See for example D. Paré et al., Impact of spontaneous synaptic activity on the resting properties of cat neocortical pyramidal neurons in vivo, J. Neurophysiol, 1998; Morales et al., J. Neurophys. 1987. The present results add the consequences of electrical coupling via mixed synapses to the established shunting by chemical synapses. The abstract and discussion should better reflect this state of understanding.

10. Line 102: "This potential shunting effect led us to examine the impact of larger electrical components of mixed PSPs on their own chemical components. In auditory and vestibular primary afferent circuits, electrical components propagate depolarizations in neighboring mixed synapses, enhancing chemical transmission at simultaneously active synapses (Alcami and Pereda, 2019)." A better reference for this phenomenon is Pereda et al., 2004 Brain Research Reviews (see Figure 4 and related text). Also, one could mention that this phenomenon was also found in invertebrates (Liu et al., Nature Comm., 2017).

11. Line 129: “Remarkably, dual electrical and chemical transmission has been observed between Ia primary sensory afferents and spinal motor neurons in adult cats (Curtis et al., 1979; Decima and Goldberg, 1976; Werman and Carlen, 1976), the origin circuit for the size principle.” These references did not show “dual electrical and chemical transmission between Ia primary sensory afferents and spinal motor neurons”, but rather the antidromic influence of motor neuron activity on presynaptic afferents. Coupling was discussed as a possible explanation along with other possibilities such as electrical field effects. Also, the Werman paper describes voltage-dependent properties of the Ia-evoked EPSP that deviates from their presumed reversal potential, but does not demonstrate its origin in electrical transmission. The presence of mixed synapses in the spinal cord of mammals was unequivocally demonstrated in Rash et al., PNAS, 1996.

12. Line 135: “The use of electrical synapses likely expands neuronal integrative properties …” Possibly cite prior demonstrations of this, e.g. Galarreta and Hestrin (2001), and Alcami and Pereda (2019).

13. It is worth considering the impact of cell morphology on the observations. For example a large dendritic tree can speed up the time constant (Rall, 1957 and 1962). The location of synaptic input also affects the time constant of postsynaptic potential (Rall, 1967). So the difference in the time constant of EPSP in Figure 2 could also arise from different locations of synapses along the dendritic tree.

Details:

14. Title suggestion: “Gap junction COUPLING impacts…”

15. Line 7. “Neuronal excitability is dictated by input resistance, which HAS MAINLY BEEN attributed to membrane leak channels.”

16. Line 10, “Moreover, differences in membrane time constants and temporal summation support greater densities of leak channels in larger neurons.”: Meaning of “support”? Maybe “suggest” or “imply”?

17. Line 13, “Stronger electrical synapses further prioritize their own…”: Perhaps replace “electrical synapses” with “electrical coupling”. Strong electrical coupling can be caused by stronger synapses or more synapses, something the present study cannot disambiguate. Check for similar instances elsewhere in the text. Also the reference for “further” here is unclear.

18. Line 14, “enhancing synchronous electrical inputs by lateral excitation”. Should this read “enhancing synchronous chemical transmission by lateral excitation”? The results and discussion talk about lateral excitation in the context of mixed electrical/chemical synapses.

19. Line 16, “that synaptic inputs not only contend with spinal neuron excitability, they contribute to it”: Meaning of “contend with”? “Synaptic input” is generally understood as “input current”. So synaptic input contributes to excitation, not to excitability.

20. Line 32, “motor neurons are still recruited by size thanks to the contribution of resistance to …” Does this argument assume that all the upstream neurons are active simultaneously? Is that justified?

21. Line 55, “…, it contributes to it.” Reference for the two “it”s?

22. Line 69, “not as predictive”: as what?

23. Line 70, “input resistance is not predictive of spinal interneuron recruitment order …”: This needs a bit more explanation for the non-expert, for example how this order relates to swimming frequency.

24. Line 79: “but that synaptic inputs contribute to input resistance …”. The results only show correlation, not causation, so “contribute” is not appropriate here.

25. Line 90: “If stronger electrical synapses contribute to leak current, we would expect…” This logic is unclear. As elsewhere the argument depends on whether the electrical synapses are onto or between target neurons. This reasoning would benefit from an equivalent circuit.

26. Line 92: “if soma size was the major source of leak current”. Perhaps better “major determinant”.

27. Line 314: “Before statistical analysis all data were tested for normality.” What was the outcome?

---

## [Author Response]

The following suggestions for revision include: improvements in readability; clarification of experimental results; the inclusion of some controls; and an expanded discussion.

We are grateful for the positive and rigorous review and the valuable suggestions for improvement. In response, we have completely re-written the manuscript to improve readability, clarify the experimental results and expand the discussion. We also include new simulation experiments to validate our experimental observations. We provide a point-by-point response below.

General readability and figures:1. Please spell out early on whether you are considering electrical synapses onto the motor neurons, or among the motor neurons, or both. The question arises already in the abstract. Obviously these scenarios predict different consequences.

This is a very important point. We are talking about electrical synapses ‘onto’ motor neurons, which can serve as conduits to report coupling electrophysiologically ‘among’ motor neurons at rest, assuming they are in the same target pool. The manuscript has been substantially revised to take this head on. We hope the revisions to the text and the additional simulation experiments now make this clearer.

2. Please include one or more equivalent circuit diagrams (ligand-gated conductances, driving forces, electrical coupling) to illustrate the postulated consequences of electrical synapses in different parts of the circuit.

We now include simulations of conductances, driving forces and coupling that better illustrate the consequences of coupling in the circuit.

3. Figure 1a: This is not helpful. Hard to spot any differences between the left and right panel. What is the significance of some leak channels being green and others white? Why are there no electrical junctions between neurons anywhere?

This figure panel has been removed.

4. Please justify the use of log-log plots in the figures? Is there a theoretical basis for the linear data fits on log-log scales?

A log scale has been used historically with regression lines to illustrate trends related to input resistance. We now state in the methods (L391) that semi-log plots and linear fits were used to illustrate significant trends in the data per Burke (1968).

Clarification of results:5. Line 82: Explain better how you separated electrical and chemical components of transmission. Is it assumed that the electrical input (membrane voltage) is the same whether or not the chemical transmission fails? If so is that justified? An equivalent circuit diagram might help. Also was the separation validated by pharmacology? Blockers are only mentioned in connection with Figure 3.

We have included more details on how we separated the fast electrical and chemical components of mixed transmission (L106), which are distinguished by their failure rates, latencies and pharmacology, per our previous publication (Figure 4a-c in Menelaou and McLean, 2019). The fast components are separated by about 1 ms, due to the delay for chemical transmission, so peak values of the fast electrical component are unlikely to be impacted significantly. This is also consistent with a lack of impact of NBQX or NMDA on the fast electrical component (Figure 4b in Menelaou and McLean, 2019).

6. Line 105: What is “reliability”? What motivates this measure? Why does it have units of mV? Explain how this measure should depend on the biophysical parameters of electrical/chemical transmission. Presumably the effects will differ depending on whether the electrical synapses are onto or among the target neurons. Without such explanation it is hard to follow the claim in line 108.

We apologize for the confusion. Reliability is the inverse of failure rate. We have now changed this to the more conventional ‘failure rate’ to avoid confusion. Since failure rates are linked to levels of presynaptic depolarization, we thought it would good to report. We interpreted this to mean that presynaptic terminals of large mixed synapses were more depolarized and thus more likely to release neurotransmitter, as proposed by Liu et al., 2017 and Pereda et al., 2004. We now make this more explicit in L123. More recent work has shown that injecting hyperpolarizing or depolarizing current in motor neurons can impact the amplitude of PSPs from excitatory interneurons, which is in line with this observation (Song et al., 2016). We now include this citation in L125. The units was a typo, thanks for catching this.

7. Cell types: In all of the analyses, various cell types are collapsed together, raising the possibility that the observed trend arises from particular cell types that differ from the others. To exclude this possibility, the authors could colour-code each data point in the scatter plots based on the cell type (or the combination of the cell types for Figure 2 and 3). For statistical analyses, the authors could examine the contribution of both cell-type (or combination of cell types) and explanatory variable (input resistance for Figure 1, 2 and 3c, the amplitude of PSP for Figure 2a) at the same time using a generalized linear model. The introduction could say more about the cell types examined in this study (type 1 and 2 V2a cells and their differences and similarities). The discussion could address how the distribution of electrical synapses among different cell types might affect spinal circuits.

We have now separated analysis based on cell type and updated Figure 1 with schematics to better outline the different cell types. We now include simulations address how the distribution of electrical synapses impacts spinal circuits.

8. Electrical synapses prioritize their own signals: One of the major claims of the paper is that “stronger electrical synapses further prioritize their own signals by shunting other inputs and enhancing synchronous electrical inputs by lateral excitation”. This claim seems to require further analysis. To directly demonstrate that electrical synapses shunt other inputs, one might show that other inputs get stronger after blocking the electrical synapse. Furthermore, it is confusing that the authors rely on Menelaou and McLean 2019 to support the claim of shunting because the connections examined in this paper come from the neurons (type I and II V2a neurons) that are active at the same time (during fast swim cycles). It would be more appropriate to examine how these electrical synapses impact the inputs from the neurons active during slow swim cycles.

We have backed off this claim. Instead, we focus on whether electrical synapses could provide the shunt normally attributed to leak channels at rest. Our simulation now takes on this issue without relying on citations to Menelaou and McLean, 2019. We agree that shunting would not occur between neurons active at the same time and have modified the text to clarify we are assessing the circuit at rest, not during activity.

Relation to prior work:9. Line 15: “Our findings challenge the view that leak channels alone dictate input resistance and membrane time constant….” In addition to leak channels, it is well known that synaptic transmission affects input resistance and membrane time constant, in particular for inhibitory synapses. See for example D. Paré et al., Impact of spontaneous synaptic activity on the resting properties of cat neocortical pyramidal neurons in vivo, J. Neurophysiol, 1998; Morales et al., J. Neurophys. 1987. The present results add the consequences of electrical coupling via mixed synapses to the established shunting by chemical synapses. The abstract and discussion should better reflect this state of understanding.

During the re-write, this language has been edited out and we now include a sentence in L24 that includes references to synaptic activity and voltage-dependent conductances to input resistance.

10. Line 102: “This potential shunting effect led us to examine the impact of larger electrical components of mixed PSPs on their own chemical components. In auditory and vestibular primary afferent circuits, electrical components propagate depolarizations in neighboring mixed synapses, enhancing chemical transmission at simultaneously active synapses (Alcami and Pereda, 2019).” A better reference for this phenomenon is Pereda et al., 2004 Brain Research Reviews (see Figure 4 and related text). Also, one could mention that this phenomenon was also found in invertebrates (Liu et al., Nature Comm., 2017).

We have updated the references accordingly in L125.

11. Line 129: "Remarkably, dual electrical and chemical transmission has been observed between Ia primary sensory afferents and spinal motor neurons in adult cats (Curtis et al., 1979; Decima and Goldberg, 1976; Werman and Carlen, 1976), the origin circuit for the size principle." These references did not show "dual electrical and chemical transmission between Ia primary sensory afferents and spinal motor neurons", but rather the antidromic influence of motor neuron activity on presynaptic afferents. Coupling was discussed as a possible explanation along with other possibilities such as electrical field effects. Also, the Werman paper describes voltage-dependent properties of the Ia-evoked EPSP that deviates from their presumed reversal potential, but does not demonstrate its origin in electrical transmission. The presence of mixed synapses in the spinal cord of mammals was unequivocally demonstrated in Rash et al., PNAS, 1996.

We have updated the references accordingly in the revised discussion.

12. Line 135: "The use of electrical synapses likely expands neuronal integrative properties …" Possibly cite prior demonstrations of this, e.g. Galarreta and Hestrin (2001), and Alcami and Pereda (2019).

Done.

13. It is worth considering the impact of cell morphology on the observations. For example a large dendritic tree can speed up the time constant (Rall, 1957 and 1962). The location of synaptic input also affects the time constant of postsynaptic potential (Rall, 1967). So the difference in the time constant of EPSP in Figure 2 could also arise from different locations of synapses along the dendritic tree.

We now include a statement to this effect in L184, suggesting that our observations could also contribute to those attributed to compartmentalization in the past. We note that our model was able to achieve the same phenomena with neurons modeled as spheres. In the methods L415 we mention that compartments will likely enhance any filtering properties observed in the model.

15. Line 7. "Neuronal excitability is dictated by input resistance, which HAS MAINLY BEEN attributed to membrane leak channels."

During the re-write, this line was removed from the abstract. However the point is now made in the introduction in L24, including other contributors to input resistance.

16. Line 10, "Moreover, differences in membrane time constants and temporal summation support greater densities of leak channels in larger neurons.": Meaning of "support"? Maybe "suggest" or "imply"?

During the re-write, this line was edited out. We do not use support in this capacity anywhere now.

17. Line 13, "Stronger electrical synapses further prioritize their own…": Perhaps replace "electrical synapses" with "electrical coupling". Strong electrical coupling can be caused by stronger synapses or more synapses, something the present study cannot disambiguate. Check for similar instances elsewhere in the text. Also the reference for "further" here is unclear.

During the re-write, this sentence was removed. We now use ‘higher-density’ electrical synapses instead of stronger, where appropriate.

18. Line 14, "enhancing synchronous electrical inputs by lateral excitation". Should this read "enhancing synchronous chemical transmission by lateral excitation"? The results and discussion talk about lateral excitation in the context of mixed electrical/chemical synapses.

Yes, this is meant to refer to the chemical component of mixed synapses. We apologize for the lack of clarity here. The sentence in L123 has been modified to avoid confusion and cites references to mixed electrical/chemical synapses.

19. Line 16, "that synaptic inputs not only contend with spinal neuron excitability, they contribute to it": Meaning of "contend with"? "Synaptic input" is generally understood as "input current". So synaptic input contributes to excitation, not to excitability.

During the re-write, this line was removed.

20. Line 32, "motor neurons are still recruited by size thanks to the contribution of resistance to …" Does this argument assume that all the upstream neurons are active simultaneously? Is that justified?

During the re-write, this line was removed.

21. Line 55, "…, it contributes to it." Reference for the two "it"s?

During the re-write, this line was removed.

22. Line 69, "not as predictive": as what?

During the re-write, this line was removed.

23. Line 70, "input resistance is not predictive of spinal interneuron recruitment order …": This needs a bit more explanation for the non-expert, for example how this order relates to swimming frequency.

We apologize for the lack of clarity. During the re-write, this line and the associated experiments were removed. We have provided more details regarding ‘fictive’ escape swimming recordings in L85 and added a schematic to Figure 1.

24. Line 79: "but that synaptic inputs contribute to input resistance …". The results only show correlation, not causation, so "contribute" is not appropriate here.

This sentence has been edited during revisions.

25. Line 90: "If stronger electrical synapses contribute to leak current, we would expect…" This logic is unclear. As elsewhere the argument depends on whether the electrical synapses are onto or between target neurons. This reasoning would benefit from an equivalent circuit.

We apologize for the confusion and clarify the logic in L116. We also include a simulation that walks the reader more clearly through this reasoning and impact.

26. Line 92: "if soma size was the major source of leak current". Perhaps better "major determinant".

Changed, thanks.

27. Line 314: "Before statistical analysis all data were tested for normality." What was the outcome?

We have added the following to clarify in L428: “Before statistical analysis all data were tested for normality *and were not normally distributed, so non-parametric tests were used*.”